# Improved performance of fNIRS-BCI by stacking of deep learning-derived frequency domain features

Jamila Akhter[1]◉, Hammad Nazeer[1]◉, Noman Naseer[1]◉*, Rehan Naeem[1]◉, Karam Dad Kallu[2]◉*, Jiye Lee[2]◉, Seong Young Ko[2]◉*

1 Department of Mechatronics and Biomedical Engineering, Air University, Islamabad, Pakistan,
2 MeRIC-Lab (Medical Robotics & Intelligent Control Laboratory), School of Mechanical Engineering, Chonnam National University, Gwangju, South Korea

◉ These authors contributed equally to this work.
* noman.naseer@au.edu.pk (NN); karam@chonnam.ac.kr (KDK); sko@jnu.ac.kr (SYK)

**Data availability statement:** Ms. Maria Malik (maria.malik@au.edu.pk) is point of contact for data correspondence from Air University. The

## Abstract

The functional near-infrared spectroscopy-based brain-computer interface (fNIRS-BCI) systems recognize patterns in brain signals and generate control commands, thereby enabling individuals with motor disabilities to regain autonomy. In this study hand gripping data is acquired using fNIRS neuroimaging system, preprocessing is performed using nirsLAB and features extraction is performed using deep learning (DL) Algorithms. For feature extraction and classification stack and fft methods are proposed. Convolutional neural networks (CNN), long short-term memory (LSTM), and bidirectional long-short-term memory (Bi-LSTM) are employed to extract features. The stack method classifies these features using a stack model and the fft method enhances features by applying fast Fourier transformation which is followed by classification using a stack model. The proposed methods are applied to fNIRS signals from twenty participants engaged in a two-class hand-gripping motor activity. The classification performance of the proposed methods is compared with conventional CNN, LSTM, and Bi-LSTM algorithms and one another. The proposed fft and stack methods yield 90.11% and 87.00% classification accuracies respectively, which are significantly higher than those achieved by CNN (85.16%), LSTM (79.46%), and Bi-LSTM (81.88%) conventional algorithms. The results show that the proposed stack and fft methods can be effectively used for the classification of the two and three-class problems in fNIRS-BCI applications.

## Introduction

In recent years, to meet the growing need for communication, control methods, and advancements in human-computer interaction have given rise to the Brain-computer interface (BCI), which allows its users to control the human body independent of the output pathways from the brain, thereby enabling the control of external devices through the modulation of brain signals [1,2]. BCI systems recognize specific patterns in brain signals to generate control commands, enabling users to interact with prosthetics, computers, and other BCI systems [3].

dataset is available to her all the time in a dedicated repository/dedicated computer. All the dataset are recorded and stored in her system for backup.

**Funding:** This study was supported by Korea Institute for Advancement of Technology(KIAT) grant funded by the Korea Government(MOTIE) in the form of a grant awarded to S. Y. Ko (RS-2024-00406796, HRD Program for Industrial Innovation). The specific roles of this author are articulated in the 'author contributions' section. The funders had no role in study design, data collection and analysis, decision to publish, or preparation of the manuscript.

**Competing interests:** The authors have declared that no competing interests exist.

Recently real-time, flexible, and adaptable BCI systems such as real-time motor imagery-based lower-limb exoskeleton controllers [4], virtual rehabilitation systems [5], remote-controlled assistive devices, and activated robotic devices pose high demands for signal processing and classification techniques. Real-time BCI applications need to deal with the non-stationary and dynamic nature of the brain signals, as well as interference from various sources of noise and artifacts. A typical BCI system is composed of five distinct phases, as depicted in Fig 1. Details of each phase following the literature review are given in this section. The first phase involves collecting signals from the brain using different methods, including invasive, semi-invasive, and non-invasive neuroimaging [6–9]. Available data acquisition modalities for BCI systems are electroencephalography (EEG), functional magnetic resonance imaging (fMRI), positron emission tomography (PET) [10], single photon emission computed tomography (SPECT), and near-infrared spectroscopy (NIRS) [11,12]. These technologies use different physical principles to measure brain activity. EEG is a neuroimaging technique that has limitations when used in BCI applications due to its low spatial resolution, sensitivity to artifacts and noise, and sensitivity to head movements [3,13,14]. fMRI has a high

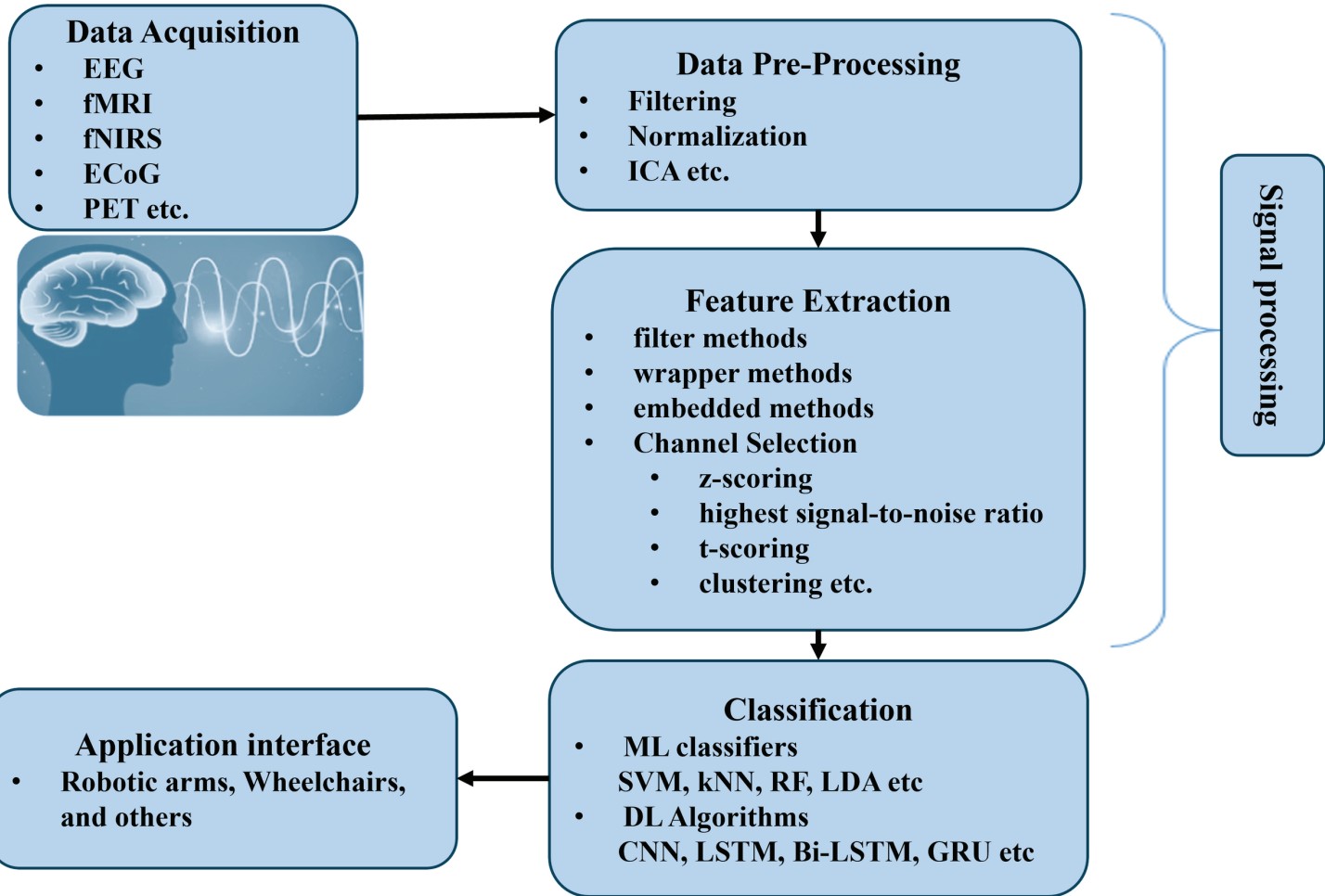

**Fig 1. Illustration of a typical BCI system.** The five phases of the BCI system are data acquisition, data pre-processing, feature extraction, classification, and application interface.

spatial resolution and can produce detailed images of the brain, but it is costly and requires subjects to lie still in a confined space [15]. On the other hand, fNIRS is relatively inexpensive, non-invasive, portable, and can measure both neural activity and blood flow, with less sensitivity to artifacts and noise [13,16–18]. In the second phase, signals are pre-processed to remove physiological artifacts that cause errors in the results [19]. In recent years, numerous techniques have been proposed such as filtering [20], normalization [21,22], and independent component analysis (ICA) [23]. In the third phase, independent and internal features are extracted [17] to overcome the redundancy of signals and for dimensionality reduction. There are several techniques for feature selection, including filter methods based on statistical properties such as variance or correlation, wrapper methods based on predictive performance, and embedded methods which are part of the learning process of a machine learning algorithm. Similarly, specific channels are identified for addressing a particular research question. Common methods for channel identification and selection include principal component analysis (PCA), z-scoring [24], the highest signal-to-noise ratio, t-scoring [25], and clustering [26]. These methods involve identifying groups of channels with similar patterns of activity or with high sensitivity to the neural activity of interest. In the fourth phase, the classification of the signals is performed based on extracted features. Machine learning (ML) algorithms, such as support vector machines (SVM), k-nearest neighbors (k-NN), linear discriminant analysis (LDA), Logistic regression (LR), Naive Bayes (NB), and random forests (RF), have been used [26–29]. Besides ML algorithms, deep learning (DL), such as convolutional neural networks (CNNs), recurrent neural networks (RNNs), long short-term memory (LSTM), and Bi-directions long short-term memory (Bi-LSTM) models have also been applied to neuroimaging signals classification [30,31]. These models are designed to learn hierarchical representations of the data, and they have been shown to achieve superior performance compared to traditional machine learning algorithms in many studies. These models have been employed in categorizing various brain activity patterns that occur during a range of cognitive and motor tasks, as well as in identifying neurological afflictions such as Alzheimer's disease and Parkinson's disease. In the last phase of BCI systems, classified signals are used to generate control commands, and to operate external devices [16]. In the neuroimaging field for BCI applications, fNIRS has become a popular technique due to its non-invasive nature, portability, and ability to measure brain activity in real-world settings [32]. In fNIRS, use a modified Beer-Lambert law (MBLL) to measure changes in cortical blood oxy- and deoxy-hemoglobin by employing two or more near-infrared wavelengths within the range of 700–1000 nm [33]. fNIRS' ability to track changes in concentration of oxy and deoxy hemoglobin signal, has a comparatively improved temporal resolution as compared to fMRI [34] and better spatial resolution as compared to EEG [35], making it more suitable for BCI applications. fNIRS-BCI signals processing and classification techniques have made significant strides yet are facing several challenges. Isolation of relevant information from the noisy biological signal and lack of standardized protocol for data acquisition and preprocessing hinder the comparability of results across similar studies. Besides that, diverse neural patterns across individuals and the adaptability of BCI systems to individual differences demand the development of universally effective algorithms. Available ML classifiers require manual feature extraction, which can be time-consuming and may miss important information. In contrast, DL algorithms can learn complex features from raw data without the need for manual feature engineering [36]. Furthermore, DL algorithms can be trained on large datasets, which can improve the accuracy and generalizability of the analysis [37]. This can be especially beneficial in clinical applications, where accurate diagnosis and treatment depend on reliable and reproducible data analysis. This study aims to enhance the classification accuracy of the fNIRS-BCI application. For this purpose, conventional DL algorithms (CNN, LSTM, and Bi-LSTM) are employed for

feature extraction, which are concatenated and then used in two proposed methods to classify the binary classification hand-gripping motor activity. In the first method extracted features from the conventional DL are used to train the proposed stack model and in the second method fast Fourier transformation (fft) technique is applied to the extracted features, then classification is performed using the proposed stack model. To evaluate the effectiveness of the proposed methods, the classification accuracies from the stack and fft methods are compared with conventional CNN, LSTM, and Bi-LSTM algorithms.

## Materials and methods

### Participants

Twenty right-handed (10 male and 10 female) participants with an age range of $25 \pm 5$ years are selected. All participants are provided with written informed consent before participating in the study. None of the participants are left-handed, have consumed caffeine less than 4 hours before the experiment, and have neurological disorders. The Institutional Review Board of Air University, Islamabad, Pakistan approved this study (approval number: AU/EA/2022/02/011), and data collection is carried out in compliance with the most recent version of the Declaration of Helsinki.

### Signal acquisition setup

Following the literature continuous-wave fNIRS imaging system (NIRSport2 acquisition system by NIRx medical technologies) [38] is utilized to acquire hand-gripping motor activity in the brain. The Optode placement shown in Fig 2 is designed to allow for imaging of the motor cortex area of the brain, following the 10–20 standard system, utilizing 8 sources and 8 detectors (16 optodes in total) [37,38]. The optodes are positioned on the motor cortex with a distance of 3 cm between each source-detector pair [39]. This arrangement resulted in the creation of a total of twenty channels (source-detector location), providing a comprehensive representation of the hemodynamic changes in response to the motor task. The sampling frequency was configured to 10.1725 Hz.

### Experimental paradigm

Initiating each experiment, the participants were instructed to take 30 seconds of rest which was then followed by 10 seconds of hand gripping and a subsequent 20 seconds of rest while remaining at rest. At the end of each experiment, a final 30 seconds of rest was provided for baseline correction of the signals. Each participant underwent 10 trials, as depicted in Fig 3. Excluding the initial (30 seconds) and final (30 seconds) resting periods, the total duration of each experiment for each subject was 300 seconds. The experimental paradigm followed is represented in Fig 3.

### Signal preprocessing

According to the literature, preprocessing of motor task (hand-gripping) is conducted using nirsLAB [16,37,38]. The nirsLAB (version: v201904_64bit) is specifically designed to process fNIRS data and offers several functionalities including the elimination of artifacts, calculation of hemodynamic response functions, and creation of brain activity maps. The data acquired from each channel of the fNIRS imaging system is in the form of light intensity from the light detectors placed on the skull surface of the motor cortex area. This light intensity is then converted into changes in the concentration of oxyhemoglobin ($\Delta C_{HbO}(t)$)

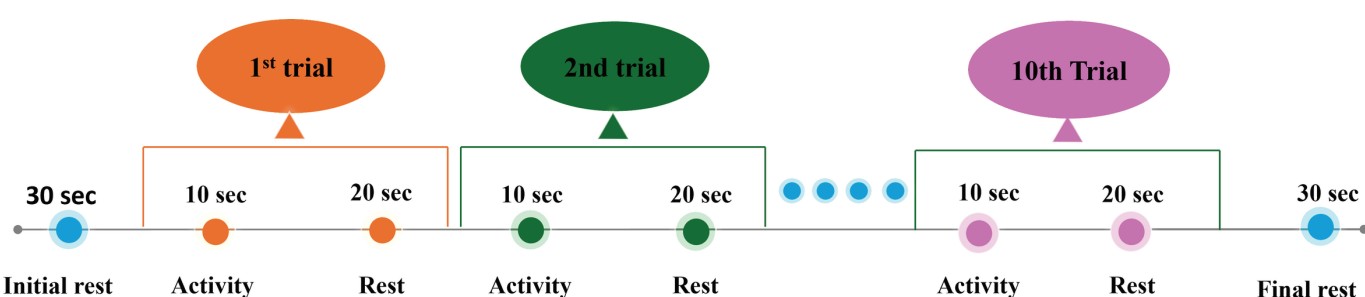

**Fig 2. Optode placement following 10–20 international system.** Sources are marked with pink circles while detectors are marked with blue circles. The setup included a total of eight sources and eight detectors, spaced 3 centimeters apart from each other, resulting in 20 channels for recording brain signals.

**Fig 3. Experimental paradigm.** Following 30 seconds of rest, each trial involved a 10-second of hand gripping activity, followed by 20 seconds of rest, again followed by 30 seconds of final rest. A total of 10 trials were conducted, resulting in a complete experimental duration of 360 seconds.

and deoxyhemoglobin ($\Delta C_{HbR}(t)$)[18,40,41]. The Modified Beer-Lambert Law (MBLL)is used to calculate the changes in hemoglobin concentrations through the application of Eq 1:

$$\left[ \begin{array}{c} \Delta C_{HbO}(t) \\ \Delta C_{HbR}(t) \end{array} \right] = \frac{\left[ \begin{array}{cc} \alpha_{HbO}(\lambda_1) & \alpha_{HbR}(\lambda_1) \\ \alpha_{HbO}(\lambda_2) & \alpha_{HbR}(\lambda_2) \end{array} \right]^{-1} \left[ \begin{array}{c} \Delta A(t;\lambda_1) \\ \Delta A(t;\lambda_2) \end{array} \right]}{l \times d} \ldots\ldots\ldots \tag{1}$$

Where,

- $\alpha_{HbO}(\lambda_1)$, $\alpha_{HbR}(\lambda_2)$ are the extinction coefficients of HbO and HbR in $[\mu M^{-1} \cdot cm^{-1}]$.
- $\Delta C_{HbO}(t)$, $\Delta C_{HbR}(t)$ represent the concentration changes of HbO and HbR, respectively, in $[\mu M]$.
- $A(t;\lambda_1)$, $A(t;\lambda_2)$ denote the absorbance measured at time $t$ using two different wavelengths $\lambda_1$ and $\lambda_2$.
- $d$ = Differential path length factor.
- $l$ = Distance between source and detector (3 cm).

Eq 1 characterizes the hemodynamic response and allows for the determination of the relative changes in oxygenated and deoxygenated hemoglobin levels in response to the motor task. To ensure the validity of the data, a bandpass filter is applied to remove physiological and instrumental noise. The filter parameters are set following the established literature [17], with a low-pass cutoff frequency of 0.2 Hz and a high-pass cutoff frequency of 0.01 Hz. Filtered data is in the form of changes in the concentration of oxyhemoglobin (HbO), deoxyhemoglobin (HbR), and total hemoglobin (HbT). In Fig 4 filtered data from channel 1 of subject 1 is plotted.

## Features extraction and classifications

The classification of filtered fNIRS signal is performed following the proposed methods, graphically represented in Fig 5. Conventional CNN, LSTM, and Bi-LSTM models are used to extract the features that are utilized in the two proposed methods.

- In the stack method, the extracted features are utilized as inputs to train a proposed stacked algorithm for classification.
- In the fft method, extracted features from conventional DL models are transformed into the frequency domain using fft. The transformed features are then used as inputs to train a proposed stacked algorithm for the classification.

### Feature extraction

The following is the description of the conventional DL models used for spatial and temporal feature extraction (summarizing critical information about brain activity and enabling effective analysis). Adam optimizer with a 0.0001 learning rate and categorical cross-entropy loss function is used to train these conventional DL algorithms. CNN model incorporates two 1D convolutional layers (64 neurons, 32 batch size, and kernel size 2×2), a max-pooling layer (1D pooling with pool size = 2), batch normalization, a flattened layer, dropout (0.2), dense layer (32 neurons, ReLU activation), and an output dense layer (2 neurons, ReLU activation). This composition enables the network to extract intricate patterns from data, as local temporal and dependencies and spatial features from the dropout layer. 32 features are extracted from 1D-CNN. LSTM and Bi-LSTM models are comprised of two layers of LSTM and Bi-LSTM

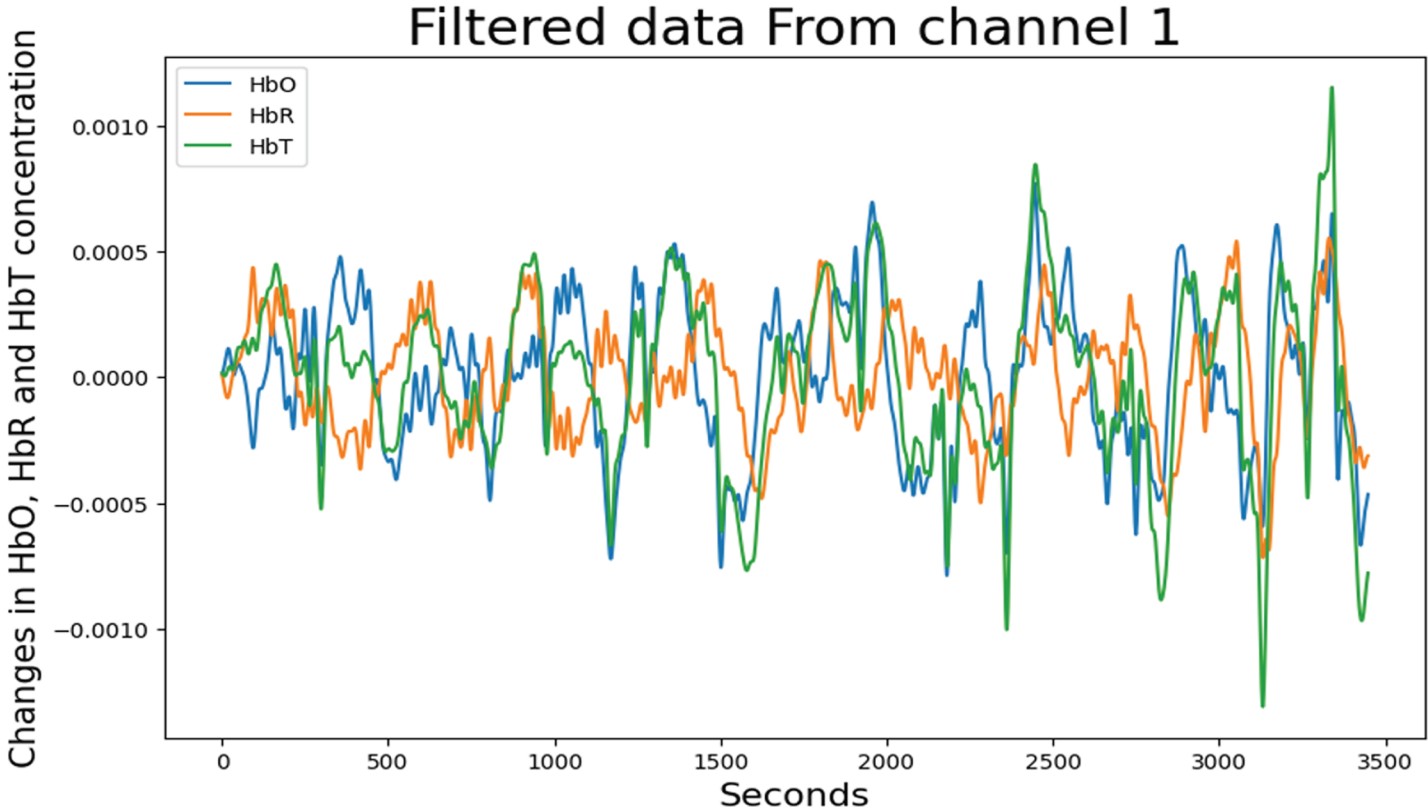

**Fig 4. Filtered data from Channel 1 of Subject 1, showing HbO, HbR, and HBT concentration changes.**

respectively (64 neurons), augmented with auxiliary layers including max-pooling (1D pooling with pool size = 2), batch normalization, flattening, dropout (0.2), dense layer (32 neurons, ReLU activation), and an output dense layer (2 neurons, ReLU activation). LSTM layers effectively capture long-range dependencies as temporal feature , enabling the network to grasp complex relationships in the data. Similarly, the utilization of two Bi-LSTM layers introduces bidirectional context, allowing the network to consider both past and future information for a more comprehensive understanding of the sequence. By this configuration of the layers in LSTM and Bi-LSTM algorithm, 32 features are extracted from each algorithm. The inclusion of auxiliary layers, such as max pooling for dimensionality reduction, batch normalization for stability, and dropout for regularization, collectively enhances the model's robustness. A total of 96 features extracted from these DL algorithms (32 from each) are used for the classification of hand-gripping activity. Once the features are extracted, these are split into test (20%) and train set (80%) to train and test the classifiers for the classification.

## Classification

In the proposed stack method, the stack model with an input layer (input is extracted features from the dropout layers of conventional DL algorithms), followed by two dense layers (64 neurons, ReLU activation in each layer) and an output dense layer (2 neurons, ReLU activation function) are used. In the proposed fft method, extracted features from the conventional DL models are transformed using fft transformation technique and then are classified using

# Proposed Methodology

**Fig 5. Schematic representation: In the proposed stack method extracted features are used to train the proposed stacked model.** In the proposed fft method extracted features are transformed using fft transformation, and then used to train the proposed stacked model for classification. Classified signals are used to generate commands to control prosthetic hand-gripping control.

the proposed stack model. The stacking approach allows for a comprehensive evaluation of the effectiveness of the extracted features in the classification of hand-gripping motor activity.

## Command generation

Acquired fNIRS signals from the brain are classified using the proposed methods and hand-open and hand-close commands generated to control the movement of the prosthetic hand gripping.

## Results

In this section, the effectiveness of the proposed methods described earlier is evaluated. The assessment of the proposed methods is in terms of model losses, accuracies, and confusion matrix. Where the model losses, accuracies, and confusion matrix of proposed stack and fft methods are compared with conventional DL algorithms.

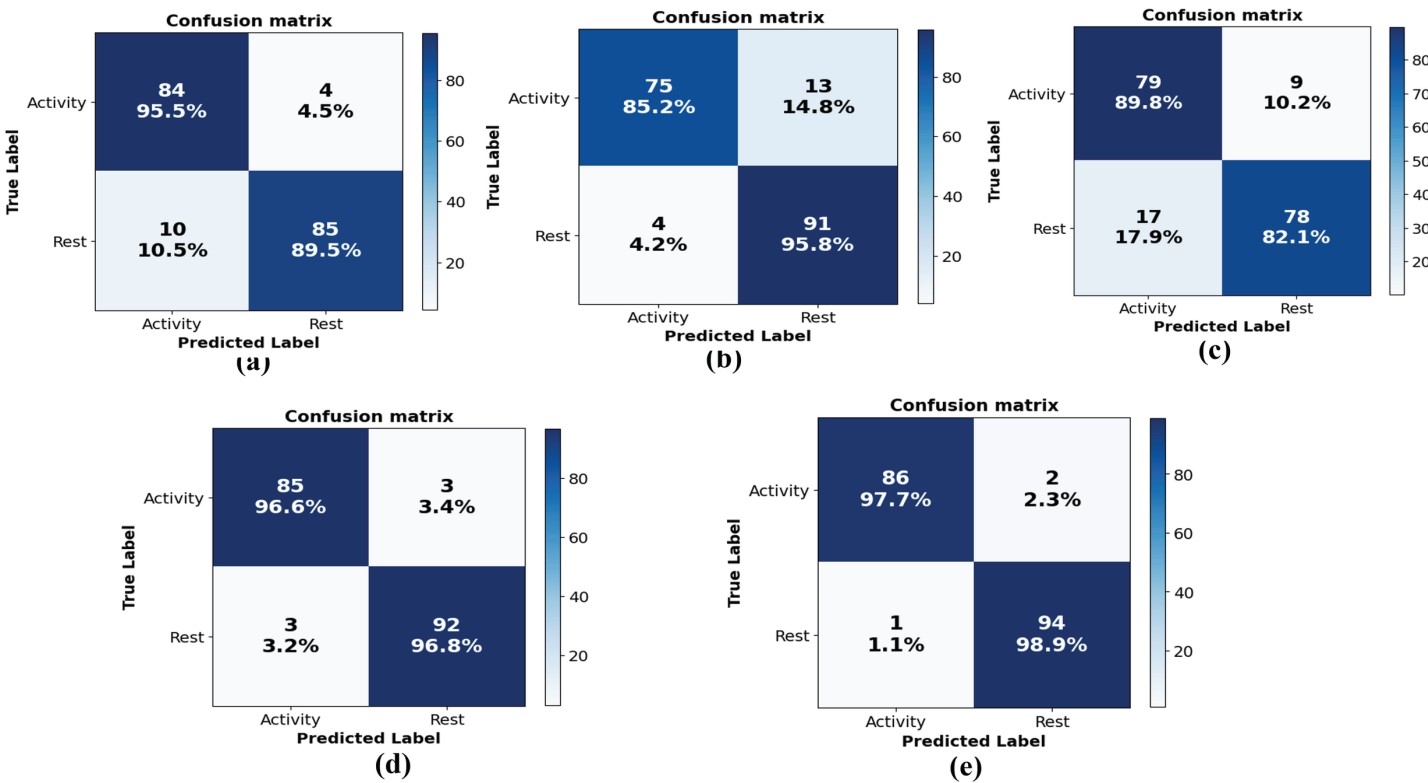

**Fig 6. Performance evaluation in terms of the confusion matrix.** Proposed stack and proposed fft methods are compared with conventional DL (CNN, LSTM, Bi-LSTM) models' performance. (a) CNN model confusion matrix. (b) LSTM model confusion matrix. (c) Bi-LSTM model confusion matrix. (d) Stack method confusion matrix. (e) fft method confusion matrix.

## Performance in terms of the confusion matrix

The confusion matrix in Fig 6a shows that the CNN model correctly predicts 84 samples of activity out of 88 samples and 85 samples of rest out of 95. Fig 6b shows that the LSTM model correctly predicts 75 samples of activity out of 88 samples and 91 samples of rest out of 95. Fig 6c shows that the Bi-LSTM model correctly predicts 79 samples of activity out of 88 samples and 78 samples of rest out of 95. Fig 6d shows that the stack method correctly predicts 85 samples of activity out of 88 samples and 92 samples of rest out of 95. Fig 6e shows that the fft method correctly predicts 86 samples of activity out of 88 samples and 94 samples of rest out of 95. In comparison amongst the proposed methods, fft method provides the highest percentage of correct predictions for both activity and rest samples. This demonstrates the effectiveness of transforming the features into the frequency domain and using them to train the proposed stacked model for prediction. A high rate of true prediction depicted in the confusion matrix implies improvement in the precision, recall, and accuracy of the proposed methods for real-time BCI applications.

## Performance in terms of model losses

The performance of conventional DL and the proposed stack models is assessed by observing the changes in both the testing and training loss over the course of several training epochs. The testing loss represents the model's ability to generalize and make accurate predictions on

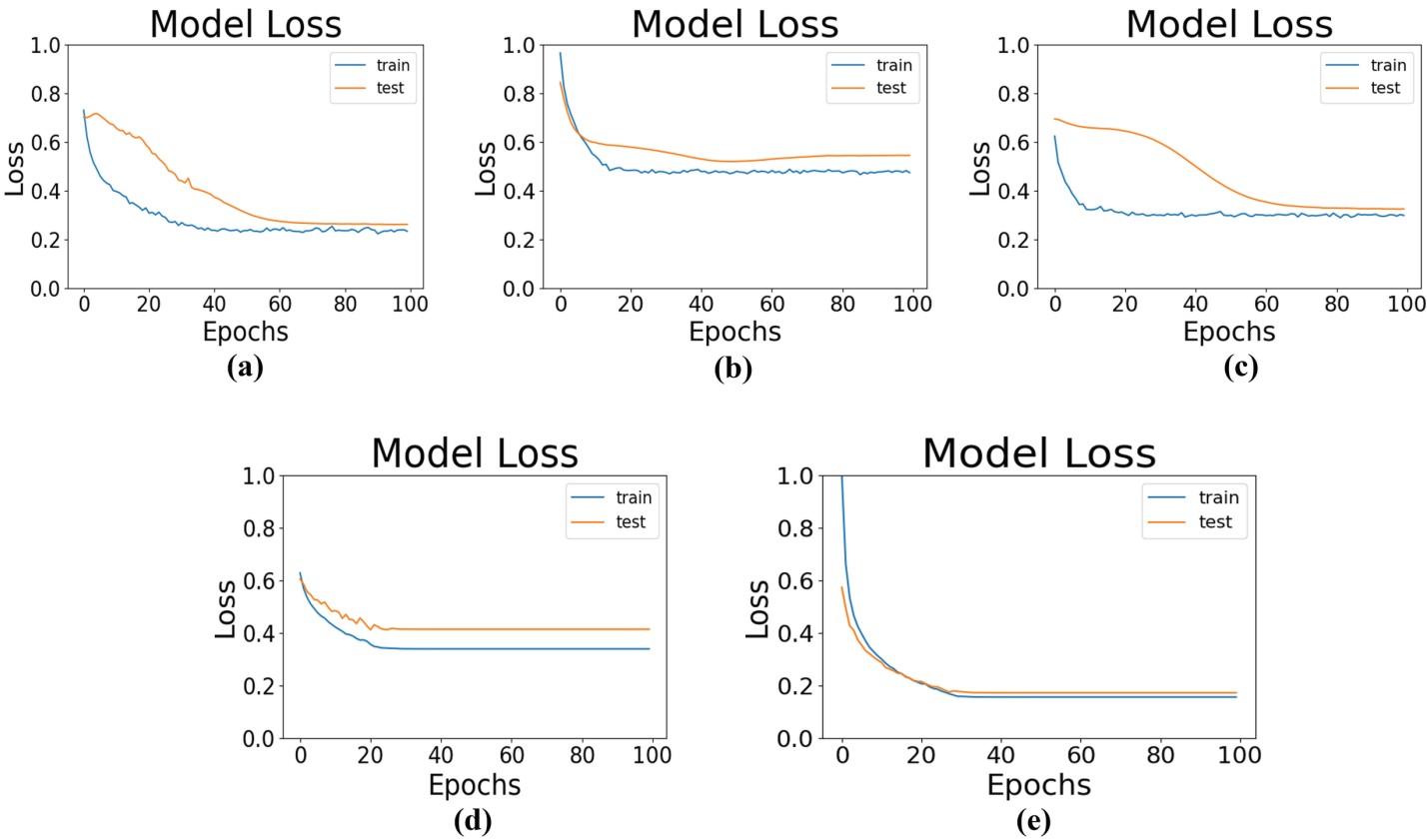

**Fig 7. Training and testing loss history of the proposed stack and proposed fft methods is compared with the conventional DL models.** (a) CNN model value losses during test and train (b) LSTM model value losses during test and train. (c) Bi-LSTM model value losses during test and train. (d) Stack method value losses during test and train (e) fft method value losses during test and train.

unseen data, while the training loss signifies the model's ability to fit the training data. These losses are commonly expressed as mean squared error or cross-entropy loss and are calculated after each epoch. In the proposed stack and fft methods, as depicted in Fig 7, the testing and training losses are low compared to the conventional DL models under evaluation. A decrease in value losses during model training has improved the accuracy of the stack model used for fft and stack method.

## Performance in terms of model training and testing accuracy

The performance in terms of model training and testing accuracy over different training epochs represents the percentage of correctly classified samples. In this study, the model testing and training accuracy of the proposed fft and stack methods is compared to that of other conventional DL models under consideration. As shown in Fig 8, the accuracy of the proposed methods improved with an increasing number of epochs, reaching its maximum value after a certain number of training iterations, indicating that in the proposed methods stacking model can effectively learn from the training data.

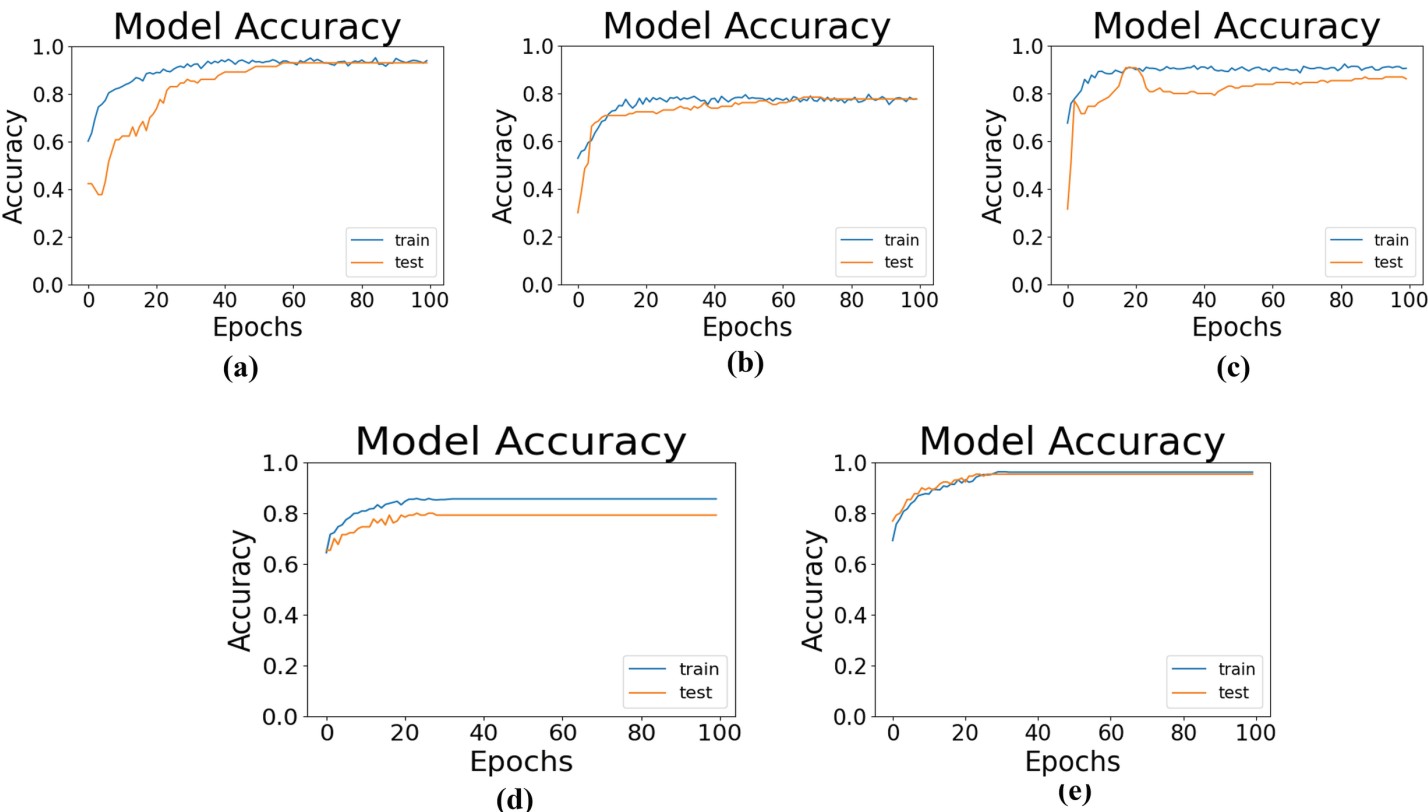

**Fig 8. Training and testing accuracy of the proposed stack and proposed fft methods are compared with conventional DL models.** (a) CNN model value accuracy during test and train (b) LSTM model value accuracy during test and train. (c) Bi-LSTM model value accuracy during test and train. (d) Stack method value accuracy during test and train. (e) fft method value accuracy during test and train.

## Performance in terms of accuracy

Classification accuracies of the proposed methods and conventional DL model are compared, in Table 1 subject-wise classification accuracies are given. These table values provide a comprehensive evaluation of the models' performance and allow for a direct comparison between the different methods. Fig 9 depicts the average accuracy comparison of proposed methods and conventional DL models. Classification accuracy of $90.11 \pm 3.65\%$ and $87.00 \pm 3.53\%$ for the HbO dataset is achieved using the proposed fft and stack method respectively. Proposed methods perform better in comparison to the classification accuracies of $85.16 \pm 7.12\%$, $79.46 \pm 3.48\%$, and $81.88 \pm 5.89\%$ using the conventional CNN, LSTM, and Bi-LSTM algorithms respectively. F1 score values provide a comprehensive evaluation performance of various models. The proposed fft method demonstrates strong performance, achieving a balance between recall and precision. Overall results in Table 2 underscore the importance of the proposed fft and stack methods.

The proposed methods demonstrate better performance for HbO datasets, achieving significantly higher average classification accuracies as compared to conventional DL models. This superiority is reflected in the results of a statistical comparison, which indicates a p-value of less than 0.0125 (Bonferroni correction applied). This is further demonstrated by the low p-value obtained in the comparison between the proposed fft, and proposed stack, DL algorithms, which is less than 0.0125, demonstrated in Table 3.

**Table 1. Subject-wise classification accuracies by using CNN, LSTM, Bi-LSTM, proposed stack, and fft methods for the classification of a 2-class hand-gripping HBO-fNIRS data.**

| Subject-wise classification accuracies | | | | | |
|---|---|---|---|---|---|
| **Subjects** | **fft Method (%)** | **Stack Method (%)** | **LSTM Algo. (%)** | **Bi-LSTM Algo. (%)** | **CNN Algo. (%)** |
| Sub 1 | 93.85 | 90.46 | 83.08 | 79.23 | 92.31 |
| Sub 2 | 93.85 | 88.92 | 79.23 | 84.62 | 90.38 |
| Sub 3 | 86.15 | 84.06 | 77.69 | 76.92 | 85.38 |
| Sub 4 | 89.23 | 85.38 | 80.00 | 80.77 | 89.23 |
| Sub 5 | 90.77 | 90.77 | 78.46 | 87.69 | 92.46 |
| Sub 6 | 93.85 | 88.62 | 73.85 | 86.92 | 93.85 |
| Sub 7 | 85.38 | 86.15 | 77.69 | 78.46 | 83.08 |
| Sub 8 | 85.38 | 83.54 | 80.77 | 76.92 | 77.69 |
| Sub 9 | 96.92 | 93.08 | 83.08 | 91.54 | 87.69 |
| Sub 10 | 89.23 | 85.15 | 75.38 | 82.31 | 86.92 |
| Sub 11 | 89.23 | 87.69 | 83.08 | 85.38 | 88.77 |
| Sub 12 | 91.54 | 88.38 | 80.00 | 90.00 | 72.31 |
| Sub 13 | 89.23 | 89.23 | 82.31 | 80.77 | 94.62 |
| Sub 14 | 85.38 | 83.08 | 71.54 | 73.85 | 77.69 |
| Sub 15 | 85.38 | 84.00 | 80.00 | 80.00 | 84.62 |
| Sub 16 | 93.08 | 85.31 | 82.31 | 84.62 | 78.46 |
| Sub 17 | 93.85 | 90.00 | 80.00 | 86.15 | 89.23 |
| Sub 18 | 87.69 | 82.31 | 81.54 | 72.31 | 86.92 |
| Sub 19 | 94.62 | 93.08 | 84.62 | 70.77 | 68.46 |
| Sub 20 | 87.69 | 80.85 | 74.62 | 88.46 | 83.08 |
| **Average** | **90.11 ± 3.65** | **87.00 ± 3.53** | **79.46 ± 3.48** | **81.88 ± 5.89** | **85.16 ± 7.12** |

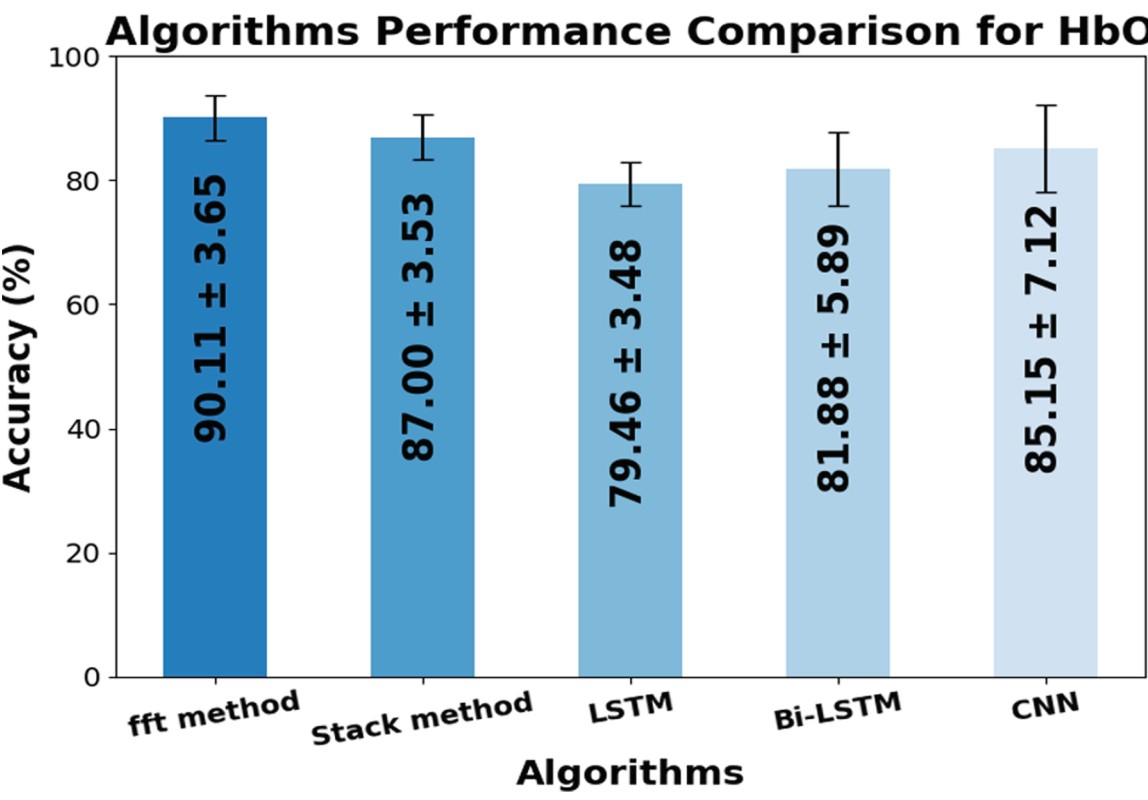

**Fig 9. Average classification accuracies for fft and stack methods, LSTM, Bi-LSTM, and CNN algorithms.**

**Table 2. Subject-wise F-1 scores by using CNN, LSTM, Bi-LSTM, proposed stack, and fft methods for the classification of a 2-class hand-gripping HBO-fNIRS data.**

| Subject-wise F1-Scores | | | | | |
|---|---|---|---|---|---|
| Subjects | CNN (%) | LSTM (%) | Bi-LSTM (%) | Stack Method (%) | fft Method (%) |
| Sub 1 | 0.9551 | 0.8910 | 0.8782 | 0.9679 | 0.8718 |
| Sub 2 | 0.8979 | 0.8315 | 0.8550 | 0.8824 | 0.9050 |
| Sub 3 | 0.7458 | 0.8987 | 0.9063 | 0.9193 | 0.9193 |
| Sub 4 | 0.8546 | 0.6263 | 0.7451 | 0.7344 | 0.9040 |
| Sub 5 | 0.8697 | 0.8916 | 0.9282 | 0.8975 | 0.9345 |
| Sub 6 | 0.8712 | 0.7575 | 0.7773 | 0.8195 | 0.9380 |
| Sub 7 | 0.6960 | 0.6798 | 0.8460 | 0.7307 | 0.8895 |
| Sub 8 | 0.7357 | 0.6200 | 0.8851 | 0.8473 | 0.9284 |
| Sub 9 | 0.8016 | 0.8805 | 0.8675 | 0.8454 | 0.9129 |
| Sub 10 | 0.8210 | 0.8743 | 0.9222 | 0.9403 | 0.9863 |
| Sub 11 | 0.6667 | 0.8354 | 0.9208 | 0.9104 | 0.8708 |
| Sub 12 | 0.7492 | 0.7522 | 0.8777 | 0.8471 | 0.9349 |
| Sub 13 | 0.6737 | 0.7347 | 0.8704 | 0.8473 | 0.8147 |
| Sub 14 | 0.8758 | 0.7742 | 0.9149 | 0.8665 | 0.9795 |
| Sub 15 | 0.6083 | 0.8271 | 0.8125 | 0.8979 | 0.8750 |
| Sub 16 | 0.7227 | 0.8503 | 0.7771 | 0.6938 | 0.8201 |
| Sub 17 | 0.7460 | 0.8789 | 0.8855 | 0.8037 | 0.9694 |
| Sub 18 | 0.6273 | 0.7112 | 0.7565 | 0.8284 | 0.8781 |
| Sub 19 | 0.7063 | 0.8721 | 0.9527 | 0.8797 | 0.9532 |
| Sub 20 | 0.5000 | 0.7892 | 0.9126 | 0.9297 | 0.9698 |
| **Average** | 0.7562 | 0.7988 | 0.8646 | 0.8545 | 0.9128 |

**Table 3. Statistical significance of the proposed method.**

| Bonferroni correction applied ($p < 0.0125$) | | | | |
|---|---|---|---|---|
| | Proposed stack model | CNN | LSTM | Bi-LSTM |
| **Proposed fft vs.** | $6.35 \times 10^{-6}$ | 0.008426795 | $1.87 \times 10^{-10}$ | $1.43 \times 10^{-6}$ |

## Validation of the proposed method

For validation purposes an open-access fNIRS dataset [42] is used, thirty participants engaged in three distinct motor movements (left and right-hand unilateral complex finger-tapping, and foot-tapping) repeated 25 times each. The initial classification accuracy achieved on this dataset was $70.4 \pm 18.4\%$ using a traditional support vector machine (SVM) classifier and leave-one-out cross-validation. However, the proposed fft method has demonstrated a remarkable enhancement, achieving an impressive average accuracy of 89% across the three-class problem. This improvement signifies a notable advancement beyond the earlier 80% accuracy mentioned in the original paper. This improvement in the accuracy using the fft method for the classification of the three-class problem as compared to the mentioned accuracy in original research on this data set demonstrates that fft and stack methods can provide a significant benefit to boosting discriminative power for complex fNIRS data through better classification performance.

## Discussion

The research community has been consistently making efforts toward enhancing the classification accuracy of state-of-the-art fNIRS-BCI systems. To achieve this goal, various methodologies have been explored including, high special and temporal resolution neuroimaging modalities for data acquisition [33], the use of advanced signal processing techniques, such

as ICA and wavelet transforms [32,43,44], general linear models [45], optimal feature selection techniques [17], [46], optimize feature selection using hybrid genetic algorithm-support vector machine (GA-SVM) [47], and optimal classification techniques [48]. To improve the performance of BCI applications use of DL algorithms [49], hybrid models that combine traditional ML algorithms with DL [16], and the integration of task-related physiological signals [50] are also highlighted. These efforts reflect a concentrated attempt to improve the performance of fNIRS-BCI systems, with the ultimate aim of providing reliable and accurate information about brain activities in real-world applications. Following the literature, a new approach is introduced to enhance the classification accuracy in fNIRS-BCI problems. The proposed methods involve feature extraction from conventional DL models and classification of these features using proposed stack and fft methods. The stack method is to train the proposed stacking model on these features from DL algorithms and in the fft method, these features are modified using fft transformation and then are used to train the stacking model. The proposed methods result in improved classification accuracy of the motor activity. The effectiveness of the proposed methods is evaluated by comparison with the traditional DL algorithm's performance. A similar gait analysis study performed using statistical features selection, and ML algorithms for classification using the different combinations of features has gained a maximum accuracy of up to 86.7% [51]. An artificial neural network (ANN) achieved 78% accuracy in classifying six-arm actions using features computed from IIR-filtered data samples [52]. The study's findings [16], show that DL models achieved a maximum accuracy of 88.5% for classifying the two-class problem, whereas the proposed methods in this study have achieved an accuracy of $90.11 \pm 3.65\%$ and $87.00 \pm 3.53\%$ using fft and stack method respectively. This study offers a new perspective in the field of fNIRS-BCI by merging the strengths of features from DL, fft transformation of these features, and the stack model. The better performance of the proposed fft method is because of the fft transformation of extracted features from the DL models and proposed stack model. This can be particularly useful in fNIRS studies where changes in brain activity are associated with specific frequencies, such as in the case of cortical oscillations [53]. By applying the fft transformation to the fNIRS extracted features from conventional DL, the time-domain signals can be transformed into the frequency domain, allowing for the identification of specific frequency components that may be relevant to the analysis. These frequency components are then used as features in the subsequent stacking process, potentially improving the classification or prediction performance of the model. In addition, using the fft reduces the dimensionality of the feature space by extracting only the relevant frequency components. This makes the subsequent stacking process more computationally efficient and reduces the risk of overfitting the model. As seen in Fig 9, the proposed methods exhibit higher accuracy of classification when compared with conventional DL models for the oxyhemoglobin dataset. This highlights the effectiveness of the proposed methods and concatenation of features from conventional DL models. The proposed fft and stacking methods have limitations such as loss of temporal information due to fft transformation, reduced interpretability due to stacking, and risk of overfitting. It may also be redundant if DL features already contain frequency information. The method was evaluated on two and three-class fNIRS datasets and used to control the hand-open and hand-close of the prosthetic hand-gripping control.

## Conclusion

This study aims to enhance the classification accuracy of an fNIRS-BCI system through the implementation of fft transformation of features from DL algorithms and the use of a stack model. The proposed fft method applies fft transformation of the features from CNN, LSTM,

and Bi-LSTM achieving an accuracy of 90.11 ± 3.65%, which is higher than conventional DL algorithms (CNN, LSTM, and Bi-LSTM). The average classification accuracies achieved by using the proposed fft method is significantly (p < 0.0125) higher than the LSTM, Bi-LSTM, and CNN algorithms. The results confirm the enhanced performance of the fft and stack methods over conventional LSTM, Bi-LSTM, and CNN algorithms in terms of accuracy, computational power, and precision. These methods potentially could advance the fNIRS-BCI systems application in cognitive monitoring and assistive technologies and could precise BCI-driven therapeutic and control devices. Despite promising results from fft and stack methods, there are limitations of fft method, as transformation during fft method loses some of the temporal information in brain signals and for large data sets stack model could face scalability challenges which increase computational cost. In the future integration of additional frequency-domain features and model optimizations to minimize computational load, could aid to use of these methods for real-time applications. Besides that these methods could be tested in the future for a wider range of motor and cognitive tasks.

## Acknowledgments

The authors express their gratitude to the National Centre of Robotics and Automation (NCRA), Rawalpindi, Pakistan, for furnishing the essential aid required to carry out this research. The authors would also like to express their appreciation to the volunteers who dedicated their time and cooperation to collecting data.

## Author contributions

**Conceptualization:** Jamila Akhter, Noman Naseer.

**Formal analysis:** Jamila Akhter, Hammad Nazeer, Karam Dad Kallu, Jiye Lee.

**Funding acquisition:** Karam Dad Kallu, Seong Young Ko.

**Investigation:** Jamila Akhter.

**Methodology:** Jamila Akhter.

**Resources:** Karam Dad Kallu, Seong Young Ko.

**Software:** Hammad Nazeer, Rehan Naeem, Jiye Lee.

**Supervision:** Noman Naseer.

**Validation:** Hammad Nazeer, Rehan Naeem.

**Visualization:** Rehan Naeem.

**Writing – original draft:** Jamila Akhter.

**Writing – review & editing:** Hammad Nazeer, Noman Naseer, Karam Dad Kallu, Seong Young Ko.

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
