## [Decision Letter · Decision Letter 0]

21 Oct 2024

PONE-D-24-21318Improved Performance Of fNIRs-BCI By Stacking of Deep Learning-Derived Frequency Domain FeaturesPLOS ONE

Dear Dr. Ko,

Thank you for submitting your manuscript to PLOS ONE. After careful consideration, we feel that it has merit but does not fully meet PLOS ONE’s publication criteria as it currently stands. Therefore, we invite you to submit a revised version of the manuscript that addresses the points raised during the review process.

The reviewers have provided valuable feedback, which I believe will help improve the quality of your work.

Please find their comments attached. We kindly ask you to carefully review and address all the points raised by the reviewers. When submitting your revised manuscript, ensure that you provide a detailed response to each comment, indicating the changes made or explaining why certain suggestions were not incorporated.

We look forward to receiving your revised manuscript.

We look forward to receiving your revised manuscript.

Kind regards,

Farzan Majeed Noori

Academic Editor

PLOS ONE

Journal Requirements:

1. When submitting your revision, we need you to address these additional requirements. Please ensure that your manuscript meets PLOS ONE's style requirements, including those for file naming. The PLOS ONE style templates can be found at https://journals.plos.org/plosone/s/file?id=wjVg/PLOSOne_formatting_sample_main_body.pdf and https://journals.plos.org/plosone/s/file?id=ba62/PLOSOne_formatting_sample_title_authors_affiliations.pdf 2. Please provide additional details regarding participant consent. In the ethics statement in the Methods and online submission information, please ensure that you have specified (1) whether consent was informed and (2) what type you obtained (for instance, written or verbal, and if verbal, how it was documented and witnessed). If your study included minors, state whether you obtained consent from parents or guardians. If the need for consent was waived by the ethics committee, please include this information. If you are reporting a retrospective study of medical records or archived samples, please ensure that you have discussed whether all data were fully anonymized before you accessed them and/or whether the IRB or ethics committee waived the requirement for informed consent. If patients provided informed written consent to have data from their medical records used in research, please include this information. 3. Please include a complete copy of PLOS’ questionnaire on inclusivity in global research in your revised manuscript. Our policy for research in this area aims to improve transparency in the reporting of research performed outside of researchers’ own country or community. The policy applies to researchers who have travelled to a different country to conduct research, research with Indigenous populations or their lands, and research on cultural artefacts. The questionnaire can also be requested at the journal’s discretion for any other submissions, even if these conditions are not met.  Please find more information on the policy and a link to download a blank copy of the questionnaire here: https://journals.plos.org/plosone/s/best-practices-in-research-reporting. Please upload a completed version of your questionnaire as Supporting Information when you resubmit your manuscript. 4. Thank you for stating the following financial disclosure: "This paper was supported by the National Research Foundation of Korea (NRF) grant funded by the Korea government (MSIT) (No. 2022R1A2C2008422) and by the “Regional Innovation Strategy (RIS)" through the National Research Foundation of Korea(NRF) funded by the Ministry of Education (MOE) (2021RIS-002)." Please state what role the funders took in the study.  If the funders had no role, please state: "The funders had no role in study design, data collection and analysis, decision to publish, or preparation of the manuscript." If this statement is not correct you must amend it as needed.  Please include this amended Role of Funder statement in your cover letter; we will change the online submission form on your behalf. 5. In the online submission form, you indicated that [The data underlying the results presented in the study can be made available by requesting Noman Naseer].  All PLOS journals now require all data underlying the findings described in their manuscript to be freely available to other researchers, either 1. In a public repository, 2. Within the manuscript itself, or 3. Uploaded as supplementary information.This policy applies to all data except where public deposition would breach compliance with the protocol approved by your research ethics board. If your data cannot be made publicly available for ethical or legal reasons (e.g., public availability would compromise patient privacy), please explain your reasons on resubmission and your exemption request will be escalated for approval.  6. Please include your full ethics statement in the ‘Methods’ section of your manuscript file. In your statement, please include the full name of the IRB or ethics committee who approved or waived your study, as well as whether or not you obtained informed written or verbal consent. If consent was waived for your study, please include this information in your statement as well. 7. Please include captions for your Supporting Information files at the end of your manuscript, and update any in-text citations to match accordingly. Please see our Supporting Information guidelines for more information: http://journals.plos.org/plosone/s/supporting-information.

Additional Editor Comments:

Dear Authors,

The reviewers have provided valuable feedback, which I believe will help improve the quality of your work.

Please find their comments attached. We kindly ask you to carefully review and address all the points raised by the reviewers. When submitting your revised manuscript, ensure that you provide a detailed response to each comment, indicating the changes made or explaining why certain suggestions were not incorporated.

We look forward to receiving your revised manuscript.

Reviewers' comments:

Reviewer's Responses to Questions

**Comments to the Author**

1. Is the manuscript technically sound, and do the data support the conclusions?

Reviewer #1: Yes

Reviewer #2: Partly

2. Has the statistical analysis been performed appropriately and rigorously? 

Reviewer #1: Yes

Reviewer #2: Yes

3. Have the authors made all data underlying the findings in their manuscript fully available?

Reviewer #1: Yes

Reviewer #2: Yes

4. Is the manuscript presented in an intelligible fashion and written in standard English?

Reviewer #1: Yes

Reviewer #2: Yes

5. Review Comments to the Author

Reviewer #1: The study is good. The positive aspect of this study is its innovative approach to enhancing fNIRS-BCI classification accuracy through the combination of deep learning feature extraction and advanced post-processing.

I have the following few minor requests:

1. Include a small justification for the 3cm distance between the source and the detector.

2. Justify why 10.1725Hz, this unusually precise rate ?

3. Mention the value of 'd' and values of extinction coefficients for HbO and HbR at the used wavelength

4. Include one line about the SNR response

5. Discuss data splitting strategy and overfitting response as well.

6. Revise and check for minor grammatical and English mistakes

Reviewer #2: This paper proposes an approach of stacking deep learning-derived frequency domain features demonstrating a novel strategy for enhancing classification performance. The idea seems very interesting. However, there are certain critical points which the authors need to address before this manuscript can be considered for publication:

• The abstract contains plenty of basic details and does not comprehensively address the novelty and unique outcomes of the study.

• The introduction section needs a complete rewriting as it directly begins with terminologies without leading the readers from a generalized concept to a specific one.

• The methodology section lacks attention to operational definitions and evidences. In certain cases, it simply says that literature says so, which is not an appropriate way to create an argument.

• In section 2.4 there is a need to specify the version of nirsLAB utilized for preprocessing. It would be beneficial to include details about any significant features or updates in this version that might influence the preprocessing steps.

• In Section 2.5 it should be specified whether the features discussed are spatial or temporal, as statistical features can fall into either category. Providing this clarification will improve the precision of the analysis and help readers better understand the context of the features being utilized.

• Specify the number of features extracted from each deep learning (DL) algorithm employed in the analysis. This information will enable readers to grasp the dimensionality of the processed data and understand the contribution of each algorithm to the overall feature set.

• Even for the Features extraction and channel selection section there needs to be an introductory paragraph on what they mean and imply.

• In DL algorithms there is a need to mention the loss function, activation function, and rate of learning used for this study to extract features from the DL algorithms.

• In Figure 4 compare the filtered data with unfiltered instead of only plotting filtered data set used (HbO data set).

• For the comparison of the results from the DL algorithms and proposed method it is suggested to do precision and/or F1-scores comparisons as well.

• There needs to be an elaboration on what the implication would be when this methodology affects the performance based on the confusion matrix and model losses.

• The validation section of the work has not been explained at all. It needs a thorough look with inclusion of which methodology was followed, along with evidence through citations.

• Lastly, while rewriting, the authors need to be elaborative on how this fft method stands unique against the previous works.

• The conclusion is very simplistic. It needs to include the implication, application, limitation and future recommendations as well.

6. PLOS authors have the option to publish the peer review history of their article (what does this mean?). If published, this will include your full peer review and any attached files.

Reviewer #1: **Yes: **Jahan Zeb Gul

Reviewer #2: No

---

## [Author Response · Author response to Decision Letter 1]

5 Nov 2024

Authors would like to extend their sincere gratitude to the editor and both reviewers for taking the time to review my manuscript. The insightful feedback and constructive suggestions are greatly appreciated and have significantly contributed to improving the quality of the work. Thank you for the valuable comments.

---

## [Decision Letter · Decision Letter 1]

12 Nov 2024

Improved Performance Of fNIRs-BCI By Stacking of Deep Learning-Derived Frequency Domain Features

PONE-D-24-21318R1

Dear Dr. Ko,

We’re pleased to inform you that your manuscript has been judged scientifically suitable for publication and will be formally accepted for publication once it meets all outstanding technical requirements.

Kind regards,

Farzan Majeed Noori

Academic Editor

PLOS ONE

Additional Editor Comments (optional):

Reviewers' comments:

Reviewer's Responses to Questions

**Comments to the Author**

1. If the authors have adequately addressed your comments raised in a previous round of review and you feel that this manuscript is now acceptable for publication, you may indicate that here to bypass the “Comments to the Author” section, enter your conflict of interest statement in the “Confidential to Editor” section, and submit your "Accept" recommendation.

Reviewer #2: All comments have been addressed

2. Is the manuscript technically sound, and do the data support the conclusions?

Reviewer #2: Yes

3. Has the statistical analysis been performed appropriately and rigorously? 

Reviewer #2: Yes

4. Have the authors made all data underlying the findings in their manuscript fully available?

Reviewer #2: Yes

5. Is the manuscript presented in an intelligible fashion and written in standard English?

Reviewer #2: Yes

6. Review Comments to the Author

Reviewer #2: Authors have revised the manuscript as per the given comments. All my suggestions including data split strategy, calculations of SNR and inclusion in the manuscript is now added.

7. PLOS authors have the option to publish the peer review history of their article (what does this mean?). If published, this will include your full peer review and any attached files.

Reviewer #2: **Yes: **Asim Waris

---

## [Editor Report · Acceptance letter]

PONE-D-24-21318R1

PLOS ONE

Dear Dr. Ko,

I'm pleased to inform you that your manuscript has been deemed suitable for publication in PLOS ONE. Congratulations! Your manuscript is now being handed over to our production team.

Kind regards,

on behalf of

Dr Farzan Majeed Noori

Academic Editor

PLOS ONE